# The orphan drug dichloroacetate reduces amyloid beta-peptide production whilst promoting non-amyloidogenic proteolysis of the amyloid precursor protein

Edward T. Parkin◯*, Jessica E. Hammond◯, Lauren Owens◯, Matthew D. Hodges

Division of Biomedical and Life Sciences, Faculty of Health and Medicine, Lancaster University, Lancaster, United Kingdom

◉ These authors contributed equally to this work.

* e.parkin@lancaster.ac.uk

**Data Availability Statement:** All relevant data are within the paper and its Supporting information files.

## Abstract

The amyloid cascade hypothesis proposes that excessive accumulation of amyloid beta-peptides is the initiating event in Alzheimer's disease. These neurotoxic peptides are generated from the amyloid precursor protein via sequential cleavage by β- and γ-secretases in the 'amyloidogenic' proteolytic pathway. Alternatively, the amyloid precursor protein can be processed via the 'non-amyloidogenic' pathway which, through the action of the α-secretase <u>a</u> <u>disintegrin</u> <u>and</u> <u>metalloproteinase</u> (ADAM) 10, both precludes amyloid beta-peptide formation and has the additional benefit of generating a neuroprotective soluble amyloid precursor protein fragment, sAPPα. In the current study, we investigated whether the orphan drug, dichloroacetate, could alter amyloid precursor protein proteolysis. In SH-SY5Y neuroblastoma cells, dichloroacetate enhanced sAPPα generation whilst inhibiting β–secretase processing of endogenous amyloid precursor protein and the subsequent generation of amyloid beta-peptides. Over-expression of the amyloid precursor protein partly ablated the effect of dichloroacetate on amyloidogenic and non-amyloidogenic processing whilst over-expression of the β-secretase only ablated the effect on amyloidogenic processing. Similar enhancement of ADAM-mediated amyloid precursor protein processing by dichloroacetate was observed in unrelated cell lines and the effect was not exclusive to the amyloid precursor protein as an ADAM substrate, as indicated by dichloroacetate-enhanced proteolysis of the Notch ligand, Jagged1. Despite altering proteolysis of the amyloid precursor protein, dichloroacetate did not significantly affect the expression/activity of α-, β- or γ-secretases. In conclusion, dichloroacetate can inhibit amyloidogenic and promote non-amyloidogenic proteolysis of the amyloid precursor protein. Given the small size and blood-brain-barrier permeability of the drug, further research into its mechanism of action with respect to APP proteolysis may lead to the development of therapies for slowing the progression of Alzheimer's disease.

**Funding:** EP received funding for this research from the Bramall Foundation (https://www.bramallfoundation.org) (no grant number available). The funders had no role in study design, data collection and analysis, decision to publish, or preparation of the manuscript.

**Competing interests:** The authors have declared that no competing interests exist.

## Introduction

The amyloid cascade hypothesis [1] states that the deposition of amyloid beta (Aβ)-peptides in the brain is the central event in the pathogenesis of the neurodegenerative condition, Alzheimer's disease (AD). These neurotoxic peptides are generated from the larger amyloid precursor protein (APP) via the 'amyloidogenic' pathway in which the protein is cleaved initially by β-secretase (β-site APP cleaving enzyme 1; BACE1) to liberate an N-terminal ectodomain termed sAPPβ and a residual beta C-terminal fragment (βCTF) [2]. This latter fragment is then cleaved by a multi-subunit protease complex known as γ-secretase in which the catalytic site resides on presenilin-1 or -2 [2, 3]. In the alternative 'non-amyloidogenic' pathway, the zinc metalloproteinase, ADAM10 (a disintegrin and metalloproteinase 10), cleaves APP within the Aβ sequence, thereby precluding the formation of intact Aβ-peptides and releasing a soluble, neuroprotective, N-terminal ectodomain termed sAPPα [4] whilst leaving behind in the membrane an alpha C-terminal fragment (αCTF). ADAM 10 also mediates the proteolytic cleavage of a range of additional cell surface integral membrane proteins within their juxtamembrane region releasing a soluble protein ectodomain into the extracellular space; a process known as 'ectodomain shedding' [5].

Dichloroacetate (DCA) is used for alleviating the symptoms of lactic acidosis associated with a number of congenital mitochondrial diseases in children [6]. Over the past few decades, chronic DCA administration has been clinically tested in adults and children in a number of disease states associated with lactic acidosis including diabetes mellitus and pulmonary arterial hypertension [7–9]. The beneficial effects of the drug in these conditions stem from its ability to activate the pyruvate dehydrogenase complex (PDC) which catalyzes the irreversible oxidative phosphorylation of pyruvate to acetyl-Coenzyme A (acetyl-CoA) thereby reducing the amount of pyruvate available for conversion to lactate [10]. This activation of the PDC is achieved through the ability of DCA to inhibit the phosphorylation and associated inactivation of the complex by pyruvate dehydrogenase kinase (PDK) [10]. More recently, and perhaps more controversially, DCA has been assigned a potential role in the treatment of various cancers [11] due to its ability to reverse the 'Warburg effect' [12] whereby cancer cells are proposed to utilize cytosolic glycolysis as a means of energy generation as opposed to the more efficient mitochondrial conversion of pyruvate to acetyl-CoA and the subsequent tricarboxylic acid cycle. The activation of the PDC by DCA is thought to reinstate the latter as the main energy generating pathway in cancer cells thereby promoting mitochondrial-mediated apoptosis.

It was the putative anti-neoplastic effect of DCA that led us to initially investigate whether the drug might impact on additional pathways linked to the proliferation of cancer cells. During these studies we remarked that the ADAM-mediated shedding of the Notch ligand, Jagged1 was enhanced by DCA. As this protein was known to be shed in a cell-specific manner by both ADAM10 and ADAM17 [13, 14], we postulated that additional substrates of these enzymes might also be affected. Therefore, in the current study, we investigated whether DCA might impact on the proteolysis of the amyloid precursor protein. When SH-SY5Y neuroblastoma cells were treated with DCA a dramatic increase in the generation of sAPPα was observed consistent with more of the APP holoprotein being cleaved via the non-amyloidogenic pathway. Conversely, the generation of sAPPβ by β-secretase activity was severely impaired with a concomitant reduction in the subsequent generation of both $A\beta_{1-40}$ and $A\beta_{1-42}$. Notably, these effects were distinct from any effects that DCA had on culture cell number; in this respect the drug exhibited an anti-proliferative rather than cytotoxic effect. APP over-expression in SH-SY5Y cells partly ablated the effects of DCA on both the non-amyloidogenic and amyloidogenic processing of the protein. In contrast, the over-expression of BACE1 in SH-SY5Y cells only partly ablated the effects of DCA on amyloidogenic but not non-

amyloidogenic APP processing. We were also able to demonstrate DCA-enhanced shedding of APP in two unrelated cell lines along with that of an unrelated ADAM substrate, Jagged1, indicating that the effect is not cell- or substrate-specific. Finally, we show that DCA did not regulate the expression of the secretases ADAM10, BACE1 or presenilin-1 or ADAM10 and BACE1 activity. These data suggest that, as DCA is a small molecule able to traverse the blood-brain-barrier, further research into its mechanism of action with respect to APP proteolysis may lead to the development of therapies for slowing the progression of Alzheimer's disease.

## Materials and methods

### Materials

The generation of the $APP_{695}$ and BACE1 constructs in the mammalian expression vector pIR-EShyg (Clontech-Takara Bio Europe, Saint-Germain-en-Laye, France) along with their expression and characterisation have been reported previously [15]. Anti-actin monoclonal (Cat. No. A4700), anti-BACE1 rabbit polyclonal (Cat. No. SAB2100200), anti-ADAM10 rabbit polyclonal (Cat. No. AB19026) and anti-APP C-terminal rabbit polyclonal (Cat. No. A8717) antibodies were from Sigma-Aldrich Company Ltd. (Poole, U.K.). Anti-APP 6E10 monoclonal (Cat. No. AB2564653) and anti-sAPPβ rabbit polyclonal (Cat. No. AB2564769) antibodies were from Biolegend (San Diego, U.S.A.). Anti-Jagged1 C-terminal goat polyclonal antibody (Cat. No. SC6011) was from Santa Cruz Biotechnology Inc. (California, U.S.A.) and anti-Jagged1 N-terminal (Cat. No. AF1277) antibody was from R & D Systems Ltd. (Abingdon, U.K.). All other materials, unless otherwise stated, were purchased form Sigma-Aldrich Company Ltd. (Poole, U.K.).

### Cell culture

Cell culture reagents were purchased from Scientific Laboratory Supplies (Nottingham, U.K.) and Thermofisher Scientific (Waltham, U.S.A.). The human colon cancer cell line SW480, HEK293 and human neuroblastoma SH-SY5Y cells were cultured in Dulbecco's modified Eagle's medium supplemented with 25 mM glucose, 4 mM L-glutamine, 10% (vol/vol) foetal bovine serum (FBS), penicillin (50 U/mL) and streptomycin (50 μg/mL). Cells were maintained at 37˚C in 5% $CO_2$ in air.

### Stable DNA transfections

Plasmids (8 μg) were linearized using AhdI before being subjected to ethanol precipitation and subsequent introduction into SH-SY5Y cells by electroporation. Recombinant cells were selected using 150 μg/ml of Hygromycin B (Invitrogen, Paisley, U.K.).

### Treatment of cells and protein extraction

SH-SY5Y cells were treated with 0, 10 or 20 mM final concentrations of DCA either at the point of seeding or once the cells had reached confluency. In the former case, the growth medium and DCA were replaced every two days until the control cells reached confluence. At this point, the growth medium was removed and cells were washed *in situ* with 10 mL of Ultra-MEM (Scientific Laboratory Supplies, Nottingham, U.K.) before adding a fresh 10 mL of UltraMEM containing the required DCA concentration and culturing for a further 24 h. Cells grown to confluence before treating with DCA were similarly washed with UltraMEM and then cultured in the same medium with or without DCA for a further 24 h. Note that a 24 h period was chosen as UltraMEM is a low serum medium (facilitating protein analysis by immunoblotting without the distortion of gels by large amounts of albumin derived from FBS

in complete medium) and, therefore, maintenance of cells in this medium becomes increasingly poor after 24 h. Following the final 24 h incubations, medium was harvested, centrifuged at 10,000 $g$ for 10 min to remove cell debris, and then equal volumes were concentrated 50-fold using Amicon Ultra-4 centrifugal filter units (Merck Millipore, Watford, U.K.). For analysis of cell-associated proteins, cells were washed with phosphate-buffered saline (PBS; 20 mM $Na_2HPO_4$, 2 mM $NaH_2PO_4$, 0.15 M NaCl, pH 7.4) and scraped from the flasks into fresh PBS (10 mL). Following centrifugation at 500 $g$ for 5 min, cell pellets were lysed in 0.1 M Tris, 150 mM NaCl, 1% (vol/vol) Triton X-100, 0.1% (vol/vol) Nonidet P-40, pH 7.4 containing a protease inhibitor cocktail (Sigma-Aldrich Company Ltd., Poole, U.K.).

## Protein assay, sodium dodecyl sulphate polyacrylamide gel electrophoresis (SDS-PAGE) and immunoblot analysis

Protein levels in cell lysates were quantified using bicinchoninic acid [16] in a microtitre plate with bovine serum albumin as a standard. Equal quantities of lysate protein and equal volumes of concentrated conditioned medium samples were resolved by SDS-PAGE using 5–20% polyacrylamide gradient gels and transferred to Immobilon P polyvinylidene difluoride (PVDF) membranes [17] before blocking in 5% (w/v) powdered milk in PBS containing 0.1% (vol/vol) Tween 20 (PBS-Tween) and incubating with primary antibody. Bound antibody was detected using peroxidase-conjugated secondary antibodies (Sigma-Aldrich Company Ltd, Poole, U.K. and R&D Systems Ltd., Abingdon, U.K.) in conjunction with enhanced chemiluminescence detection reagents (Perbio Science Ltd., Cramlington, U.K.). For re-probing with anti-actin antibody, the original antibodies were stripped from membranes by heating at 50˚C for 30 min in 100 mM β-mercaptoethanol, 2% (wt/vol) SDS, 62.5 mM Tris pH 6.7 with occasional agitation. The membranes were then washed twice for 10 min in PBS-Tween before blocking and re-probing with primary antibody as described above.

For the resolution of APP C-terminal fragments, samples were run on 16% Tris/tricine gels before transferring to 0.2 micron nitrocellulose (Fisher Scientific, Loughborough, U.K.) and subsequently boiling for 5 min in PBS. Membranes were then further processed as described above.

## Cell viability assays

For trypan blue assays, cells were harvested by trypsinisation and an aliquot from the resultant resuspended cell pellet was mixed with an equal volume of 0.4% (wt/vol) trypan blue solution and loaded onto a haemocytometer. Cell counts of live cells were obtained and an average of four squares taken. For MTS (methanethiosulfonate) assays, cells were incubated with CellTiter 96$^®$ AQueous One Cell Proliferation Assay solution (Promega, Wisconsin, U.S.A.) for 2 h at 37˚C. Absorbance readings at 490 nm were then taken using a Victor$^2$ 1420 microplate reader (Perkin Elmer, Waltham, U.S.A.). Note that the two types of viability assays were performed separately from each other using independent cultures.

## Aβ-peptide quantification

Aβ-peptides in unconcentrated conditioned medium samples were quantified using the Mesoscale Discovery (MSD) platform. $Aβ_{1-40}$ and $Aβ_{1-42}$ were measured using the V-Plex Aβ peptide panel (6E10) kit according to the manufacturer's instructions (MSD, Maryland U.S.A.).

## Real-time quantitative polymerase chain reaction (qPCR)

RNA was extracted from SH-SY5Y cells using Trizol (Thermofisher Scientific, Waltham, U.S.A.) following the manufacturer's instructions and yield was determined by spectroscopy on a

Nanodrop 2000 (Thermofisher Scientific, Waltham, U.S.A.). cDNA for real-time qPCR analysis was synthesized using SuperScript III (Thermofisher Scientific, Waltham, U.S.A.) following the manufacturer's instructions. Typically 1000 ng of RNA was used in each synthesis reaction. Real-time qPCR analysis was performed on a CFX96 thermal cycler (Bio-Rad Laboratories, Watford, U.K.) using SYBR green Jumpstart Taq ready mix (Sigma-Aldrich Company Ltd., Poole, U.K.) and reaction conditions were as follows: initial denaturation at 95°C for 5 min, followed by 40 cycles of 94°C for 1 min, 63°C for 1 min and 68°C for 15 s. Reactions were performed in triplicate in volumes of 20 μl. The analysis was then repeated on triplicate independent biological samples. Relative expression was based on ΔΔCt methodology analysed in CFX software version 3.1 (Bio-Rad Laboratories, Watford, U.K.). Reaction primers for BACE 1 were 5'-CAGTCATCCACGGGCACTGT-3' (forward) and 5'-CTGAACTCATCGTGCA CATGGC-3' (reverse). Primers for ADAM10 were 5'-GGCTTCACAGCTCTCTGCCCA-3' (forward) and 5'-CCTGCACATTGCCCATTAATGCA-3' (reverse). Primers for presenilin-1 were 5'-GCCAGAGAGCCCTGCACTCAA-3' (forward) and 5'-GCATGGATGACCTTATAG CACC-3' (reverse). Primers for ribosomal protein L15 (RPLO) were 5'-GCAATGTTGCCA GTGTCTG-3' (forward) and 5'-GCGTTGACCTTTTCAGCAA-3' (reverse).

### Fluorimetric secretase activity assays

ADAM10 activity in cell lysates was assayed using a SensoLyte® 520 ADAM10 activity fluorimetric assay kit (Anaspec, Fremont, U.S.A.) according to the manufacturer's instructions. 5-FAM fluorescence was monitored at excitation/emission = 490 nm / 520 nm using an Infinite M200 Pro Tecan plate reader (Tecan Trading, Männedorf, Switzerland). Controls using the ADAM10 inhibitor GM6001 were incorporated and results adjusted to compensate for non-specific substrate degradation. BACE1 activity in cell lysates was measured using a Fluorometric Beta Secretase activity assay kit (Abcam, Cambridge, U.K.) according to the manufacturer's instructions. Fluorescence was monitored at excitation/emission = 335 nm / 495 nm using the Tecan plate reader described above. Controls using the β-secretase inhibitor supplied as part of the kit were incorporated and results adjusted to compensate for non-specific substrate degradation. Results from both fluorimetric assays were also adjusted to account for differences in total protein concentrations between lysate samples.

### Statistical analysis

Data are presented as means ± S.D. and were subjected to statistical analysis via one-way analysis of variance (ANOVA) with Tukey's post hoc tests (analysed using IBM SPSS software). No outliers were excluded. Levels of significance are indicated in figure legends.

## Results

### Growth of untransfected SH-SY5Y cells in the presence of DCA enhances non-amyloidogenic and inhibits amyloidogenic processing of endogenous APP

Initially, human SH-SY5Y cells (untransfected) were seeded in the absence or presence of DCA and cultured until the control flasks reached confluence (medium and DCA were replaced every two days). At this point the growth medium was replaced with UltraMEM containing the same DCA concentrations (see Materials and methods) and the cells were cultured for an additional 24 h. Trypan blue assays of cultures at this point demonstrated 25.02 ± 1.37 and 44.93 ± 7.89% decreased viable cell numbers in 10 mM and 20 mM DCA-treated cultures, respectively, relative to controls (Fig 1A). Note that few if any non-viable cells were seen in any

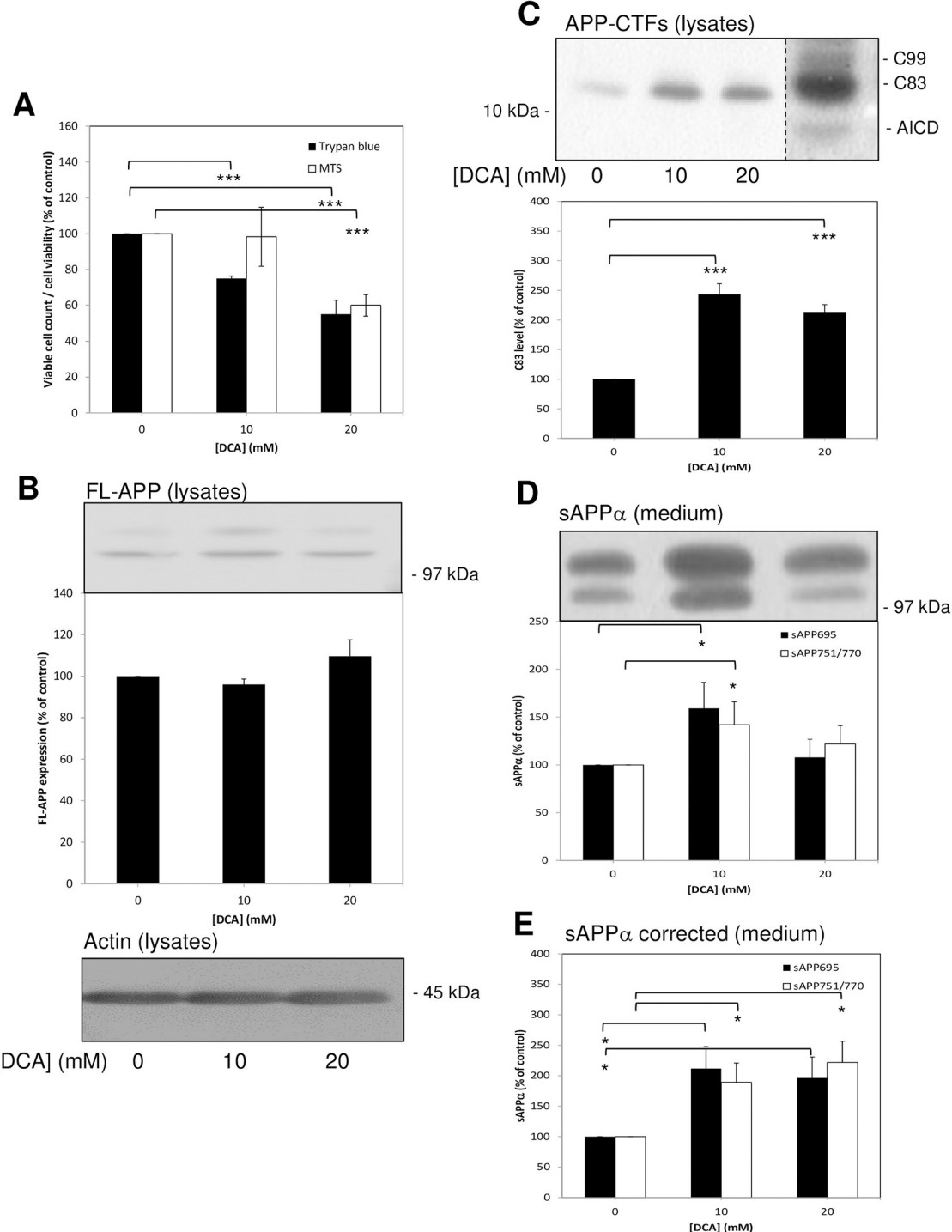

**Fig 1. Growth of untransfected SH-SY5Y cells in the presence of DCA enhances non-amyloidogenic APP processing.** Cells were seeded and cultured in the absence or presence of the indicated DCA concentrations and grown until the control cultures reached confluence (medium and DCA were replaced every two days). At this point the growth medium was replaced with UltraMEM containing the same DCA concentrations and the cells were cultured for an additional 24 h. Viability assays were then performed or cell lysate and conditioned medium samples were prepared as described in the Materials and methods section. Equal amounts of protein (lysates) or equal volumes (medium) from samples were subjected to SDS-PAGE and immunoblotting (Materials and methods). **(A)** Trypan blue and MTS cell viability assays. (**B**) Lysates were immunoblotted with anti-APP C-terminal and anti-actin antibodies. Full-length APP (FL-APP) was quantified from multiple immunoblots and the results expressed relative to control values. **(C)** Lysates were resolved on Tris/tricine gels and immunoblotted with anti-APP C-terminal antibody to detect APP C-

terminal fragments (APP-CTFs). The far right lane is a sample from APP over-expressing SH-SY5Y cells and the dashed line shows where lanes from different time exposures of the same immunoblot have been rearranged for illustrative purposes. AICD, APP intracellular domain. C83 was quantified from multiple immunoblots and the results expressed relative to control values. **(D)** Conditioned medium samples were immunoblotted with anti-APP 6E10 antibody in order to detect sAPPα. Multiple immunoblots were then quantified and results expressed relative to control values without correcting for cell number. **(E)** The results in **(D)** were corrected to account for relative changes in cell number using data from the trypan blue assays in **(A)**. All results are expressed relative to control (0 mM DCA) values and are means ± S.D. (n = 3). * = significant at $p < 0.05$; *** = significant at $p < 0.005$.

of these assays indicating a growth inhibitory rather than cytotoxic effect of DCA. Similarly, MTS cell viability assays demonstrated a 40.00 ± 6.02% decrease in cell number in the 20 mM DCA-treated cultures relative to controls although no significant decrease could be determined at a 10 mM concentration of the drug (Fig 1A).

When cell lysates prepared from cultures grown in the presence of DCA were subjected to immunoblotting using anti-APP C-terminal antibody, no significant changes were observed in levels of full-length APP (FL-APP) (Fig 1B). Equal protein loading between lysate samples was confirmed by re-probing immunoblots with anti-actin antibody (Fig 1B). Lysate samples were also resolved on Tris/tricine gels and immunoblotted with the same anti-APP C-terminal antibody in order detect APP C-terminal fragments (APP-CTFs). The results (Fig 1C) demonstrated highly significant increases specifically in the levels of α-secretase-generated C83 of 2.44 ± 0.18- and 2.14 ± 0.12-fold, respectively, in lysates prepared from 10 and 20 mM DCA-treated cells (relative to untreated controls). Levels of C99 and the APP intracellular domain (AICD) were below the limits of detection in untransfected SH-SY5Y cells.

In contrast to lysate samples, conditioned medium samples (from the final 24 h UltraMEM incubation) were resolved on gels on an equal volume (as opposed to equal protein) basis. Immunoblotting of medium samples with anti-APP 6E10 antibody revealed increases of 59.17 ± 27.12 and 42.13 ± 23.98%, respectively, in the levels sAPPα derived from $APP_{695}$ (lower band) and $APP_{751/770}$ (combined in the upper band on immunoblots) shed from 10 mM DCA-treated relative to control cells (Fig 1D). The data presented in Fig 1D were then adjusted (using the trypan blue assay data) in order to compensate for the reduction in cell number observed following DCA treatment (i.e. to provide information on the amount of protein shed per unit cell number). The resultant data (Fig 1E) revealed actual 111.70 ± 36.08 and 89.03 ± 31.89% increases, respectively, in the levels of $sAPP_{695}\alpha$ and $sAPP_{751/770}\alpha$ shed from cells treated with 10 mM DCA compared to control cells. Similarly, at 20 mM DCA, the levels of these fragments shed were increased by 96.23 ± 34.51 and 121.80 ± 34.87%, respectively, compared to the controls.

The same conditioned medium samples were then immunoblotted with anti-sAPPβ antibody and the subsequent quantification of multiple immunoblots revealed that sAPPβ generation from $APP_{695}$ was almost completely inhibited (unquantifiable in some immunoblots) in the presence of either 10 or 20 mM DCA (Fig 2A). Similarly, sAPPβ generated from $APP_{751/770}$ was reduced by 61.88 ± 12.16 and 84.82 ± 15.67%, respectively, in 10 mM and 20 mM DCA-treated cultures compared to controls. Even when these data were adjusted in order to account for decreased cell numbers following DCA treatment the results still revealed 49.30 ± 16.17 and 72.37 ± 28.52% reductions, respectively, in the levels of sAPPβ generated from $APP_{751/770}$ following 10 and 20 mM DCA treatments (Fig 2B). When unconcentrated conditioned medium samples were analysed in terms of Aβ-peptide content, the results (Fig 2C and 2D) showed that $A\beta_{1-40}$ was reduced by 69.46% (59.60% corrected for cell number) and 94.72% (90.40% corrected), respectively, following treatment with 10 and 20 mM DCA. Similarly, $A\beta_{1-42}$ levels were reduced by 58.93% (45.24% corrected) and 89.29% (80.36% corrected), respectively, at the same drug concentrations.

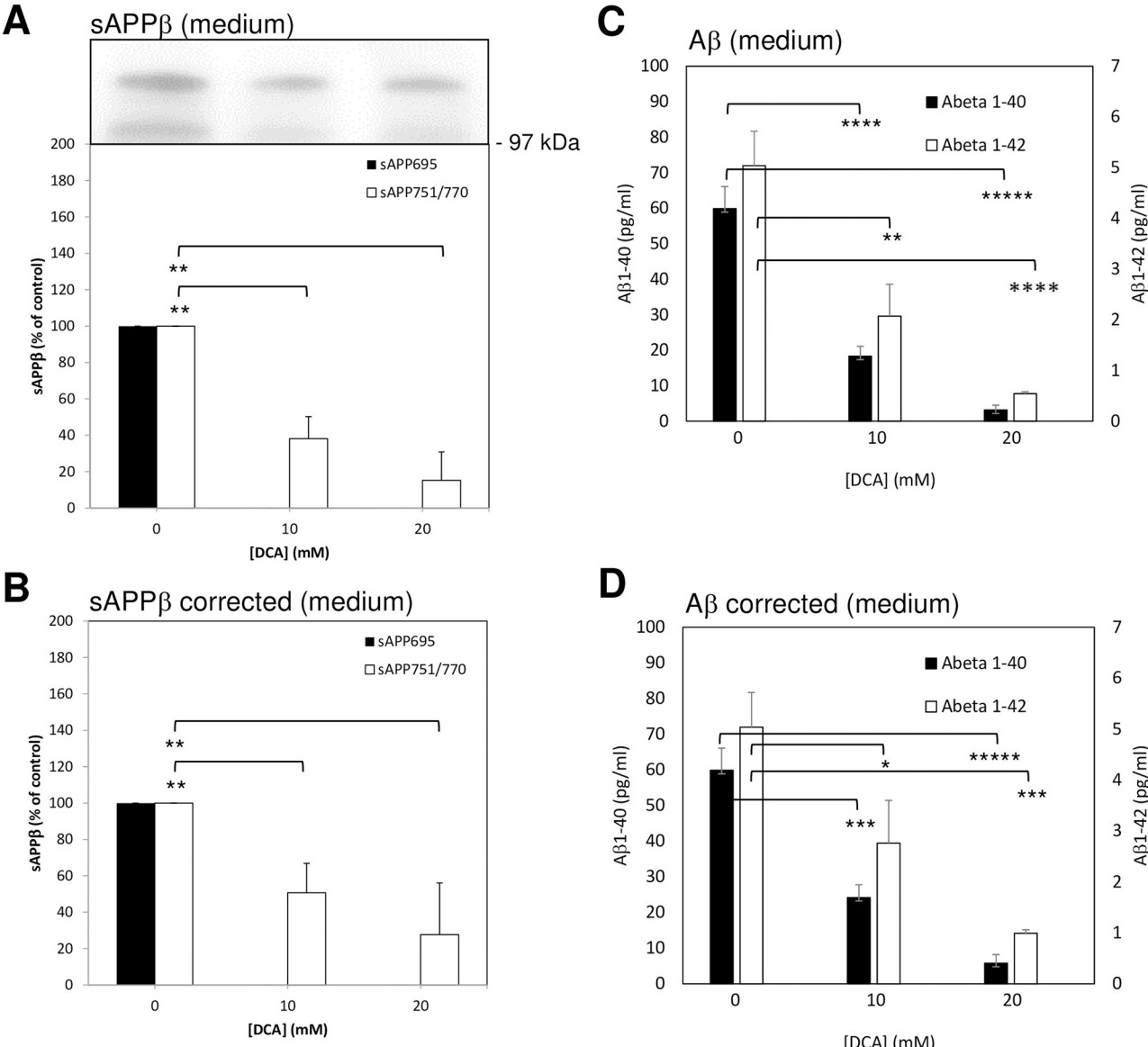

**Fig 2. Growth of untransfected SH-SY5Y cells in the presence of DCA inhibits amyloidogenic APP processing.** Cells were seeded and cultured in the presence of the indicated DCA concentrations and grown until the control cultures reached confluence (medium and DCA were replaced every two days). At this point the growth medium was replaced with UltraMEM containing the same DCA concentrations and the cells were cultured for an additional 24 h. Conditioned medium samples were either used unconcentrated (Aβ-peptide analysis) or concentrated before subjecting equal volumes to SDS-PAGE and immunoblotting (Materials and methods). **(A)** Conditioned medium samples were immunoblotted with anti-sAPPβ antibody and multiple immunoblots were quantified and results expressed relative to control values without correcting for cell number. **(B)** The results in **(A)** were corrected to account for relative changes in cell number using data from the trypan blue assays in Fig 1A. **(C)** and **(D)** Aβ-peptide quantification in conditioned medium before **(C)** and after **(D)** correcting for changes in cell number. The sAPPβ results are expressed relative to control (0 mM DCA) values whereas the Aβ-peptide results are absolute values (pg/mL). All results are means ± S.D. (n = 3). * = significant at $p < 0.05$; ** = significant at $p < 0.01$; *** = significant at $p < 0.005$; **** = significant at $p < 0.001$; ***** = significant at $p < 0.0005$.

## The effects of DCA on endogenous APP proteolysis in SH-SY5Y cells are distinct from any effects of the drug on cell number

In the preceding experiments we observed what appeared, given the lack of appreciable non-viable cell numbers, to be a growth inhibitory (as opposed to cytotoxic) effect of DCA. In

order to circumvent the need to correct results for changes in cell number, we next grew untransfected SH-SY5Y cells to confluence before replacing the growth medium with Ultra-MEM and treating them with DCA for just 24 h. Indeed, trypan blue and MTS cell viability assays showed no appreciable effects of the drug on cell number following this treatment (Fig 3A).

When cell lysates from 24 h DCA-treated cells were immunoblotted using anti-APP C-terminal antibody (Fig 3B) a slight but significant increase in APP expression was observed only in the 10 mM drug-treated cell lysates; equal protein loading between lysate samples was confirmed by re-probing immunoblots with anti-actin antibody (Fig 3B). When the conditioned medium from these cells was immunoblotted with anti-APP 6E10 antibody the results (Fig 3C) revealed increased $sAPP_{695}\alpha$ shedding from 10 mM and 20 mM DCA-treated cells but these changes could not be quantified due to the absence of a detectable signal in the samples from control cells. However, the shedding of $sAPP_{751/770}\alpha$ was quantifiable and showed $5.63 \pm 0.38$- and $6.23 \pm 0.56$-fold increases, respectively, in medium from 10 and 20 mM DCA-treated cells relative to controls.

The same conditioned medium samples were then immunoblotted with anti-sAPPβ antibody (Fig 3D) and quantification of the results revealed $79.87 \pm 13.17$ and $100\%$, respectively, decreases in the shedding of $sAPP\beta_{751/770}$ from 10 mM and 20 mM DCA-treated cells relative to controls (no signal for $sAPP\beta_{695}$ was detected in these samples). When unconcentrated conditioned medium was analysed in terms of Aβ-peptide content, the results (Fig 3E) showed that $A\beta_{1-40}$ was reduced by $33.71\%$ and $60.79\%$, respectively, in medium from 10 mM and 20 mM DCA-treated cells relative to controls. Similarly, $A\beta_{1-42}$ was reduced by $24.87\%$ and $52.13\%$ at the same drug concentrations.

Collectively, these data show that DCA is able to enhance the non-amyloidogenic and inhibit the amyloidogenic processing of endogenous APP in untransfected SH-SY5Y cells. Of note, we also tested the effect of DCA on endogenous APP processing in two additional unrelated cell lines (SW480 colon cancer and HEK293 cells) and demonstrated similar enhancements in the non-amyloidogenic processing of endogenous APP (although neither of these cell lines generate quantifiable endogenous levels of amyloidogenic processing proteolytic fragments) (S1 Fig).

## Stable over-expression of APP in SH-SY5Y cells partially ablates the effects of DCA on both non-amyloidogenic and amyloidogenic proteolysis

The results in the preceding section suggested that DCA might impair amyloidogenic APP processing through a simple reciprocal enhancement of non-amyloidogenic processing leaving less substrate available for cleavage by BACE1. We, therefore, hypothesized that saturating the ADAM-mediated shedding of APP by over-expressing the latter protein should, at least in part, ablate the effects of DCA. Consequently, we repeated the 24 h DCA treatment experiments using SH-SY5Y-$APP_{695}$ stable transfectants which dramatically over-express (approximately 50-fold) this smaller isoform of the protein [15]. No appreciable effects of DCA on cell number were observed over the treatment period (Fig 4A). Similarly, immunoblotting of cell lysates using anti-APP C-terminal antibody revealed no significant changes in the levels of FL-APP following DCA treatment (Fig 4B); equal protein loading between lysate samples was confirmed by re-probing immunoblots with anti-actin antibody (Fig 4B).

When conditioned medium samples from the same experiments were immunoblotted using the anti-APP 6E10 antibody (Fig 4C), levels of $sAPP_{695}\alpha$ shed from 10 mM and 20 mM DCA-treated cells were shown to be enhanced by only $1.83 \pm 0.16$- and $2.26 \pm 0.34$-fold, respectively, relative to controls (compare this with the $5.63 \pm 0.38$- and $6.23 \pm 0.56$-fold

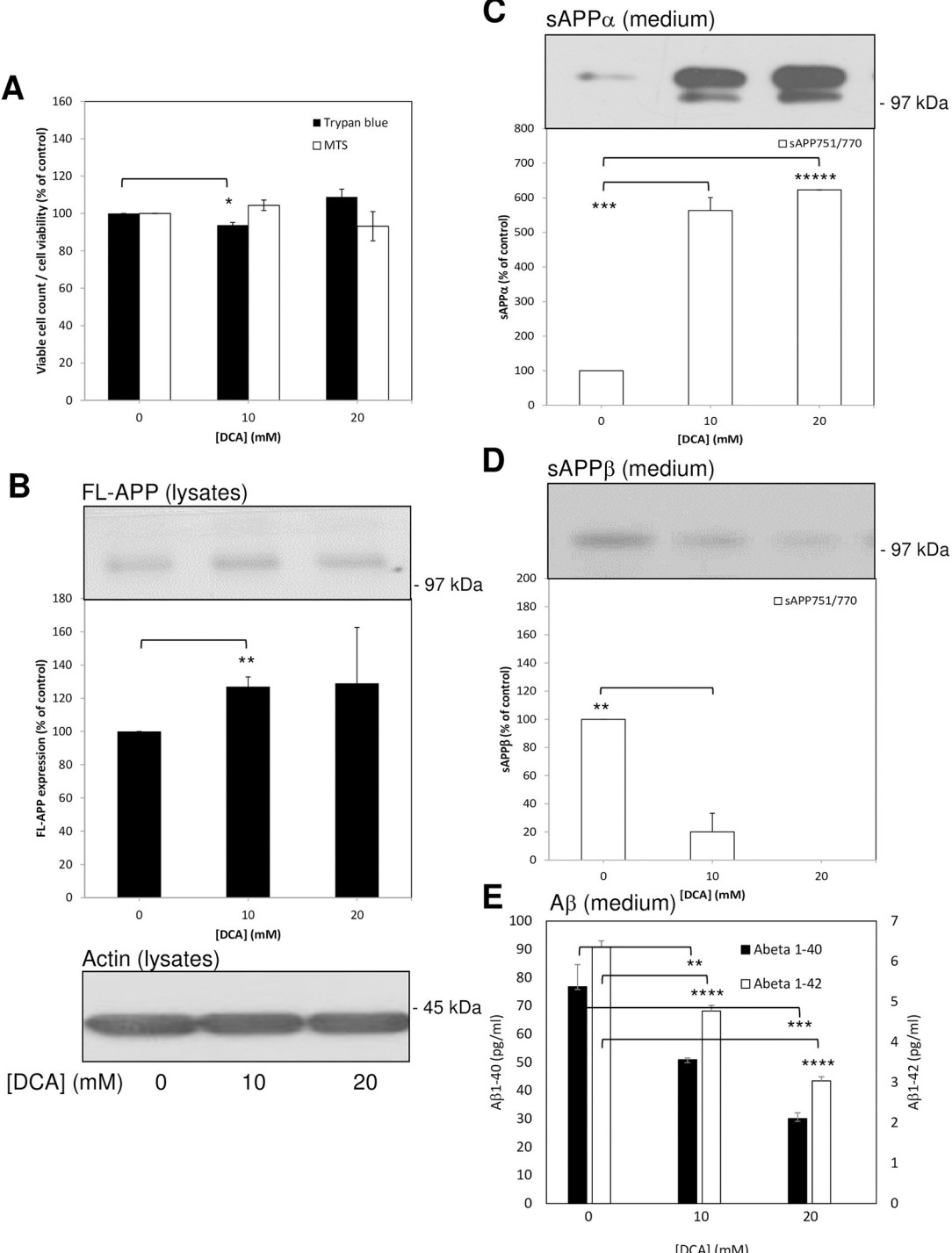

**Fig 3. Treatment of confluent untransfected SH-SY5Y cells with DCA enhances non-amyloidogenic and inhibits amyloidogenic APP processing.** Cells were grown to confluence before replacing the growth medium with UltraMEM containing the indicated DCA concentrations and culturing for an additional 24 h. Viability assays were then performed or cell lysate and conditioned medium samples were prepared as described in the Materials and methods section. Equal amounts of protein (lysates) or equal volumes (medium) from samples were subjected to SDS-PAGE and immunoblotting (Materials and methods). **(A)** Trypan blue and MTS cell viability assays. (**B**) Cell lysates were immunoblotted with anti-APP C-terminal and anti-actin antibodies. Full-length APP (FL-APP) was quantified from multiple immunoblots and the results expressed relative to control values. **(C)** and **(D)** Conditioned medium samples were immunoblotted with anti-APP 6E10 antibody in order to detect sAPPα **(C)** or anti-sAPPβ antibody **(D)**. Multiple immunoblots were then quantified and results expressed relative to control

values. **(E)** Aβ-peptide quantification in conditioned medium; results are absolute values (pg/mL). All results are means ± S.D. (n = 3). * = significant at $p < 0.05$; ** = significant at $p < 0.01$; *** = significant at $p < 0.005$; **** = significant at $p < 0.001$; ***** = significant at $p < 0.0005$.

increases observed for endogenous $sAPP_{751/770}\alpha$ using untransfected SH-SY5Y cells; Fig 3C). When the same samples were immunoblotted with the anti-sAPPβ antibody it was clear that DCA had no significant effect on the generation of this fragment by $SH-SY5Y-APP_{695}$ cells (Fig 4D). Similarly, no significant effect of 10 mM DCA on Aβ-peptide levels in conditioned medium could be detected and, at a 20 mM concentration of the drug, $A\beta_{1-40}$ and $A\beta_{1-42}$ levels were reduced by 32.97% and 25.01%, respectively (Fig 4E) (compare this with the equivalent 60.79% and 52.13% reductions observed in untransfected cells; Fig 3E).

Collectively, these data indicate that saturating the non-amyloidogenic pathway of APP proteolysis reduces the impact that DCA has on the proportion of substrate processed through this route. Concomitantly, the amount of APP substrate available for cleavage by BACE1 would not be as greatly affected by DCA as would be the case in an unsaturated system (i.e. endogenous levels of APP).

## Stable over-expression of BACE1 in SH-SY5Y cells partially ablates the effect of DCA on amyloidogenic but not non-amyloidogenic proteolysis

Assuming that DCA simply shifts competition for the APP substrate in favour of α-secretase activity, over-expression of BACE 1 might be expected to reverse this competition and, therefore, negate the effects of the drug on Aβ-peptide generation. Consequently, we repeated the 24 h DCA treatment experiments using SH-SY5Y-BACE1 stable transfectants which dramatically over-express (approximately 50-fold) the enzyme [15]. Cell viability assays demonstrated no change in cell number following a 24 h treatment with 10 mM DCA (Fig 5A). At 20 mM DCA, whereas the trypan blue assay showed no change in cell number, the MTS assay revealed a significant 29.82 ± 1.71% decrease in viability.

Next, lysates from the SH-SY5Y-BACE1 cells were immunoblotted with anti-BACE1 antibody and multiple immunoblots were quantified. The results (Fig 5B) showed an intriguing dose-related increase in BACE1 expression (1.17 ± 0.13- and 1.62 ± 0.23-fold, respectively, at 10 and 20 mM DCA). Immunoblots were re-probed with anti-actin to confirm equal protein loading (Fig 5B). In contrast, no change in levels of endogenous FL-APP were detected when the same samples were immunoblotted with anti-APP C-terminal antibody (Fig 5C); again equal lysate protein loading was confirmed by immunoblotting with anti-actin antibody (Fig 5C). When the conditioned medium from these cells was immunoblotted with anti-APP 6E10 antibody the results (Fig 5D) revealed increased $sAPP_{695}\alpha$ shedding from DCA-treated cells (5.57 ± 0.22- and 4.46 ± 0.67-fold, respectively, at 10 and 20 mM). Similarly, the shedding of $sAPP_{751/770}\alpha$ was enhanced 5.32 ± 0.28- and 5.16 ± 0.34-fold. However, when the conditioned medium was immunoblotted with the anti-sAPPβ antibody, no DCA-induced changes in secretion of this fragment could be detected (Fig 5E). Finally, quantification of Aβ-peptides in unconcentrated conditioned medium samples showed no changes in medium from 10 mM DCA-treated cells relative to controls and, at the 20 mM drug concentration, $A\beta_{1-40}$ and $A\beta_{1-42}$ levels (Fig 5F) were reduced by 33.44% and 31.11%, respectively (compare this with the equivalent 60.79% and 52.13% reductions observed in untransfected cells; Fig 3E).

Collectively, these data show that increasing BACE1 levels in cells can partly ablate the effect of DCA on amyloidogenic APP processing whilst only having a very minor impact on the ability of the drug to enhance non-amyloidogenic proteolysis.

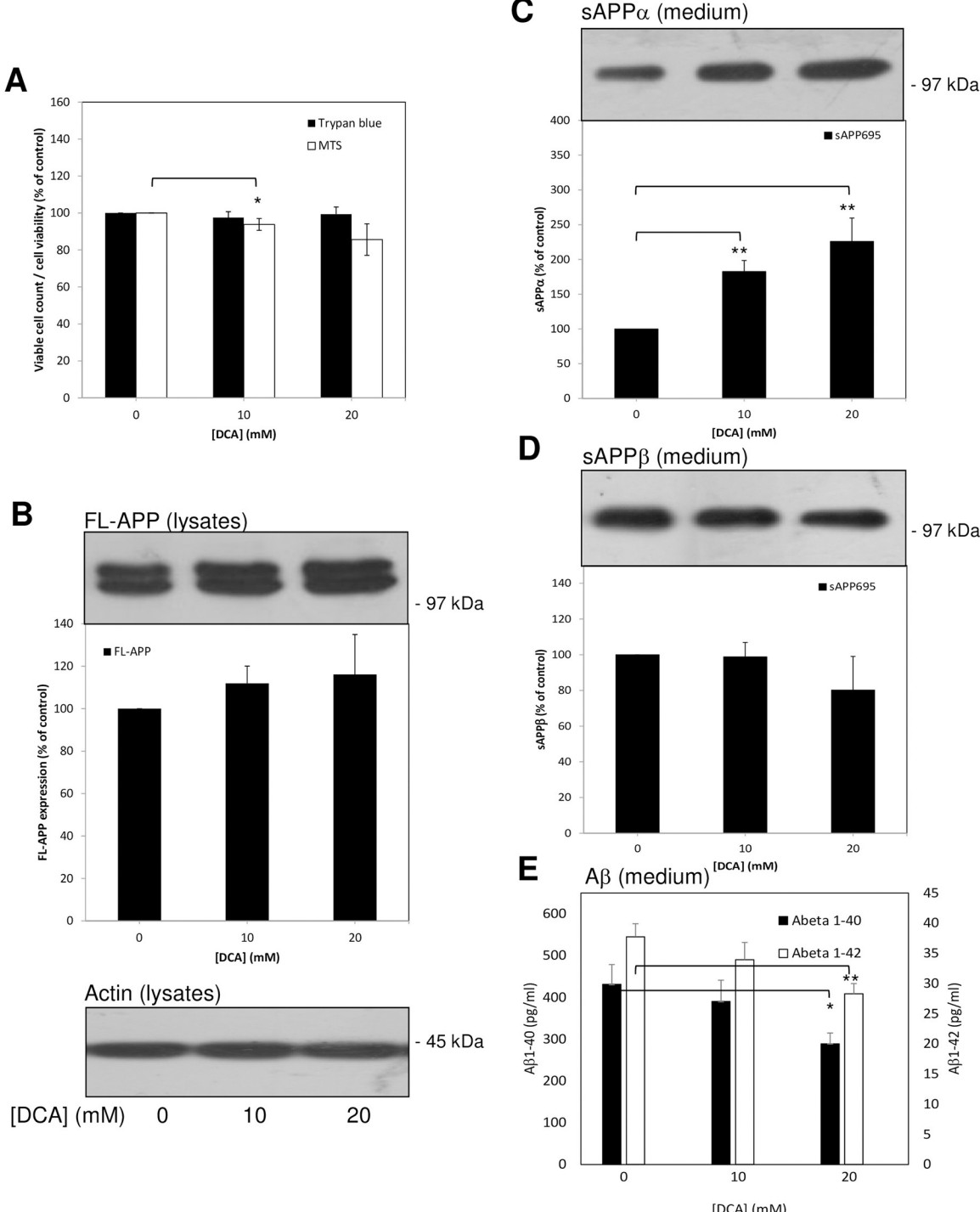

**Fig 4. Stable over-expression of APP in SH-SY5Y cells partially ablates the effects of DCA on non-amyloidogenic and amyloidogenic APP processing.** Cells were grown to confluence before replacing the growth medium with UltraMEM containing the indicated DCA concentrations and culturing for an additional 24 h. Viability assays were then performed or cell lysate and conditioned medium samples were prepared as described in the Materials and methods section. Equal amounts of protein (lysates) or equal volumes (medium) from samples were subjected to SDS-PAGE and immunoblotting (Materials and methods). **(A)** Trypan blue and MTS cell viability assays. **(B)** Cell lysates were immunoblotted with anti-APP C-terminal and anti-actin antibodies. Full-length APP (FL-APP) was quantified from multiple immunoblots and the results expressed relative to control values. **(C)** and **(D)** Conditioned medium samples were immunoblotted with anti-APP 6E10 antibody in order to detect sAPPα **(C)** or anti-sAPPβ antibody **(D)**. Multiple immunoblots were then quantified and results expressed relative to control values. **(E)** Aβ-peptide quantification in conditioned medium; results are

absolute values (pg/mL). All results are means ± S.D. (n = 3). * = significant at $p < 0.05$; ** = significant at $p < 0.01$; **** = significant at $p < 0.001$.

### DCA does not alter secretase expression or activity

As DCA appeared to promote non-amyloidogenic and inhibit amyloidogenic endogenous APP processing, we hypothesized that the drug might simply alter the expression or activity of one or more secretases. To this end, untransfected SH-SY5Y cells were grown to confluence, washed *in situ* with UltraMEM and then incubated in the same medium with or without DCA for a further 24 h. Following harvesting, cell pellets were split and used for protein and RNA extraction (Materials and methods). Immunoblotting of cell lysates using anti-ADAM10 antibody revealed no change in the expression or processing of this physiological α-secretases (Fig 6A). Furthermore, real-time qPCR performed on the RNA extracted from the same cell pellets showed no significant differences in the relative normalized expression of either ADAM10, BACE1, or presenilin-1 following DCA treatment (Fig 6B). We also conducted parallel experiments in which the resultant cell pellets were lysed and subjected to commercial fluorimetric assays for ADAM10 and BACE1 according to the manufacturer's instructions (see Materials and methods). However, the results (Fig 6C) also showed no significant effect of DCA on the catalytic activity of these two key secretases.

### The enhancement of ADAM-mediated shedding by DCA is not unique to APP

Our data show that endogenous APP shedding from SW480 cells was enhanced by DCA (S1 Fig). As these cells endogenously express the Notch ligand, Jagged1, which is also known to be shed by a member(s) of the ADAM family [13, 14], we used them to investigate whether or not the effect of DCA on protein shedding was unique to APP. SW480 cells were grown to confluence, washed *in situ* with UltraMEM, and then incubated for a further 24 h in the absence or presence of DCA. This length of incubation caused no decrease in cell viability (S1 Fig). When cell lysates from these incubations were prepared and immunoblotted with anti-Jagged1 C-terminal antibody, the results showed no significant changes in the expression level of full-length Jagged1 (Fig 7A). However, levels of the Jagged1 CTF generated by ADAM cleavage were enhanced 1.49 ± 0.26- and 1.98 ± 0.48-fold, respectively, in 10 and 20 mM DCA-treated cells compared to controls (Fig 7A); immunoblotting with anti-actin antibody confirmed equal protein loading (Fig 7B). Similarly, immunoblotting of conditioned medium from the same cultures using anti-Jagged1 N-terminal antibody (Fig 7B) showed that shedding of the Jagged1 N-terminal fragment (NTF) from cells was enhanced 2.21 ± 0.13- and 2.45 ± 0.27-fold, respectively, in cultures treated with 10 and 20 mM DCA.

### Discussion

In the current study, we report that the orphan drug dichloroacetate can simultaneously enhance non-amyloidogenic and inhibit amyloidogenic proteolysis of endogenous APP. One caveat is that plasma DCA concentrations in patients treated with the drug have previously been reported as being approximately ten-fold lower than the lowest concentration employed here [18]. However, it should be noted that, *in vitro*, higher concentrations of DCA have been shown to elicit a range of cellular effects not related to PDK inhibition e.g. effects on mitochondrial polarisation [19] and mitophagy [20]. As such, we wanted to employ drug concentrations likely to encompass a range of mechanisms and, therefore, elected to adopt higher

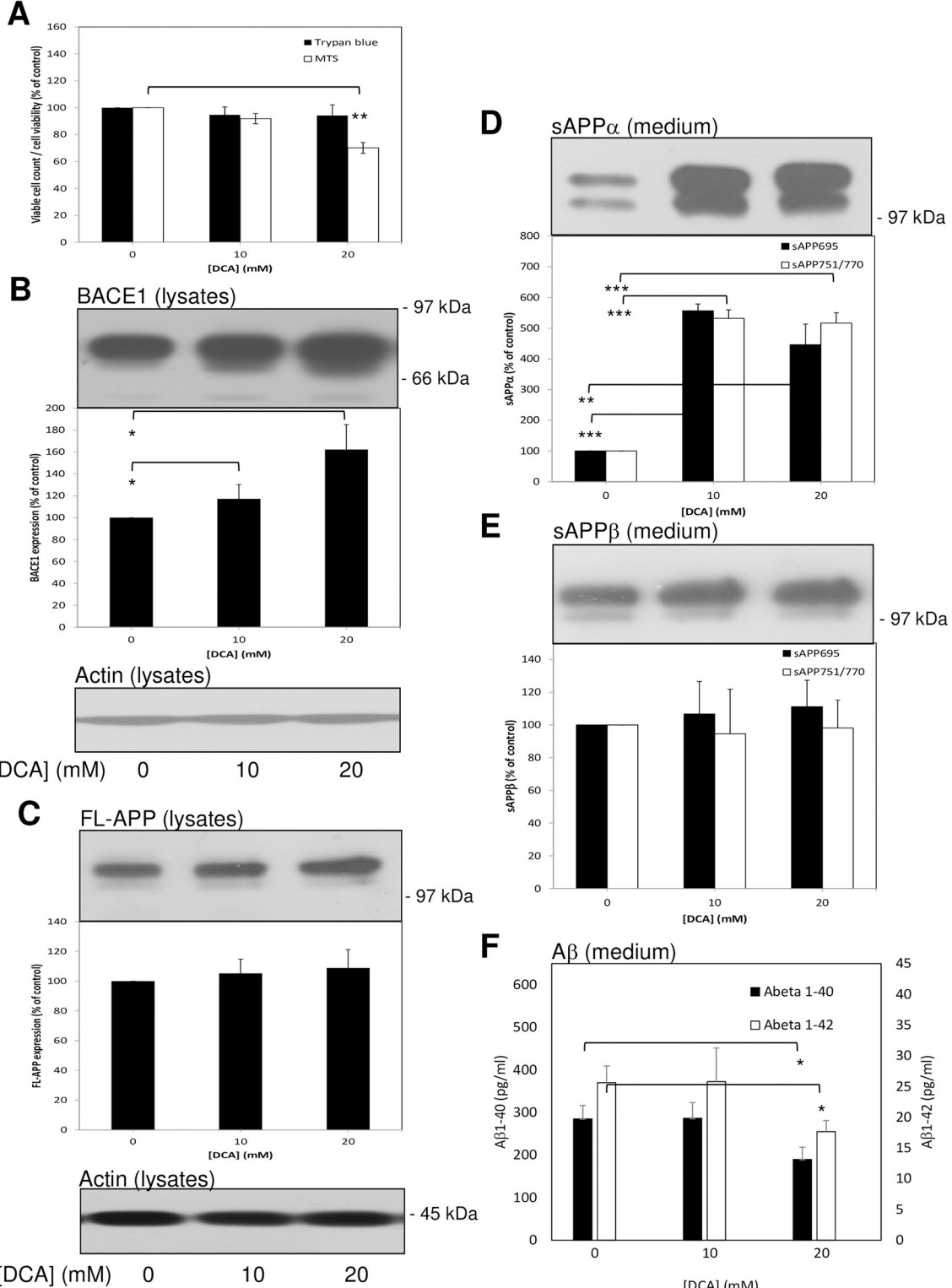

**Fig 5. Stable over-expression of BACE1 in SH-SY5Y cells partially ablates the effect of DCA on amyloidogenic but not non-amyloidogenic APP processing.** Cells were grown to confluence before replacing the growth medium with UltraMEM containing the indicated DCA concentrations and culturing for an additional 24 h. Viability assays were then performed or cell lysate and conditioned medium samples were prepared as described in the Materials and methods section. Equal amounts of protein (lysates) or equal volumes (medium) from samples were subjected to SDS-PAGE and immunoblotting (Materials and methods). **(A)** Trypan blue

and MTS cell viability assays. **(B)** Cell lysates were immunoblotted with BACE1 antibody and re-probed with anti-actin antibody. Multiple anti-BACE1 immunoblots were quantified and the results expressed relative to control values. **(C)** Cell lysates were immunoblotted with anti-APP C-terminal antibody and re-probed with anti-actin antibody. Full-length APP (FL-APP) was quantified from multiple immunoblots and the results expressed relative to control values. **(D)** and **(E)** Conditioned medium samples were immunoblotted with anti-APP 6E10 antibody in order to detect sAPPα **(D)** or anti-sAPPβ antibody **(E)**. Multiple immunoblots were then quantified and results expressed relative to control values. **(F)** Aβ-peptide quantification in conditioned medium; results are absolute values (pg/mL). All results are means ± S.D. (n = 3). * = significant at $p < 0.05$; ** = significant at $p < 0.01$; *** = significant at $p < 0.005$.

concentrations more commonly used in studies of an *in vitro* nature [21–24]. The future identification of these mechanisms with respect to the effects of DCA on APP proteolysis may lead to the development of novel therapeutics for the treatment of AD.

Initially, when untransfected SH-SY5Y cells were cultured in the presence of DCA we observed a decrease in the number of viable cells (Fig 1A) in agreement with a previous study [20]. This study used DCA concentrations up to 60 mM and the authors observed cells in suspension following drug treatment suggesting cell death. In the current study, we did not observe appreciable numbers of cells in suspension but our highest DCA concentration was only 20 mM. Furthermore, the trypan blue assay did not demonstrate appreciable numbers of non-viable cells following DCA treatment. Additionally, when cells were grown to confluence and then treated for 24 h with the drug, no differences in the number of viable cells were detected over the duration of the treatment (Fig 3A). Therefore, our data indicate that, at least at the concentrations employed in the current study, DCA was growth inhibitory rather than cytotoxic. These aspects aside, the effects of the drug on APP proteolysis were clearly distinct from changes in cell number as the former but not the latter was altered in the 24 h incubation experiments. However, DCA is clearly cytotoxic to cells derived from a range of cancers [11] and, given the neuroblastoma origin of SH-SY5Y cells, it would not be at all surprising if the drug was cytotoxic to these cells at higher concentrations. Whether the drug is cytotoxic or growth inhibitory in primary neurons remains to be tested although the fact that chronic DCA administration, whilst being associated with peripheral neuropathy, has not been shown to exert any negative cognitive effects in patients treated for mitochondrial disease, would suggest that toxicity to neurons in the brain would not be problematic despite the compound freely traversing the blood-brain-barrier [25, 26]. In fact, chronic oral administration of DCA in a mouse model of amyotrophic lateral sclerosis has been shown to improve survival and motor performance [27].

The non-amyloidogenic processing of endogenous APP was enhanced in several unrelated cell lines (SH-SY5Y, HEK293 and SW480) demonstrating that the effect of DCA in this respect was not specific to a single cell type. Furthermore, the process was enhanced in untransfected SH-SY5Y cells regardless of whether the cells were grown in the presence of DCA (Fig 1C–1E) or merely incubated with the drug for 24 h once they reached confluence (Fig 3C). In the latter experiments endogenous APP shedding was enhanced 5.63 ± 0.38- and 6.23 ± 0.56-fold, respectively, in medium from cells treated with 10 and 20 mM DCA compared to only 1.83 ± 0.16- and 2.26 ± 0.34-fold in APP over-expressing SH-SY5Y cells (Fig 4C). As the ADAM-mediated shedding of APP in the latter cell line is already massively enhanced due to increased expression of FL-APP [15], the ability of DCA to further stimulate shedding might well have been limited by the substrate saturation of ADAM activity. The fact that the ability of DCA to promote non-amyloidogenic APP processing is reduced following over-expression of the protein also questions whether AD drug action is actually best studied in animal models over-expressing APP. In this respect drugs which might effectively be used to regulate the proteolysis of APP expressed at levels more physiologically relevant to the human brain might

## A

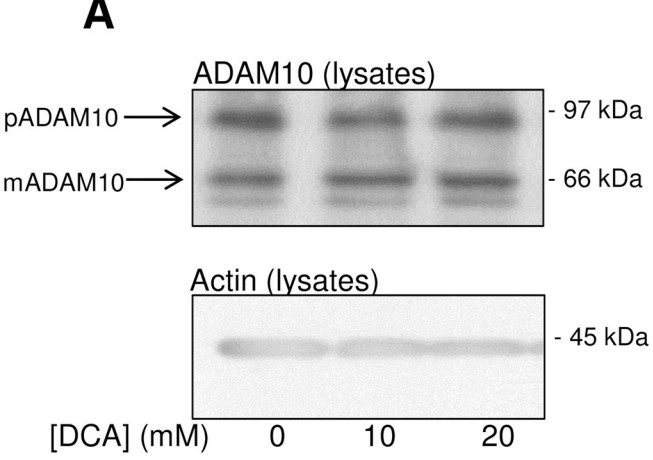

## B   C

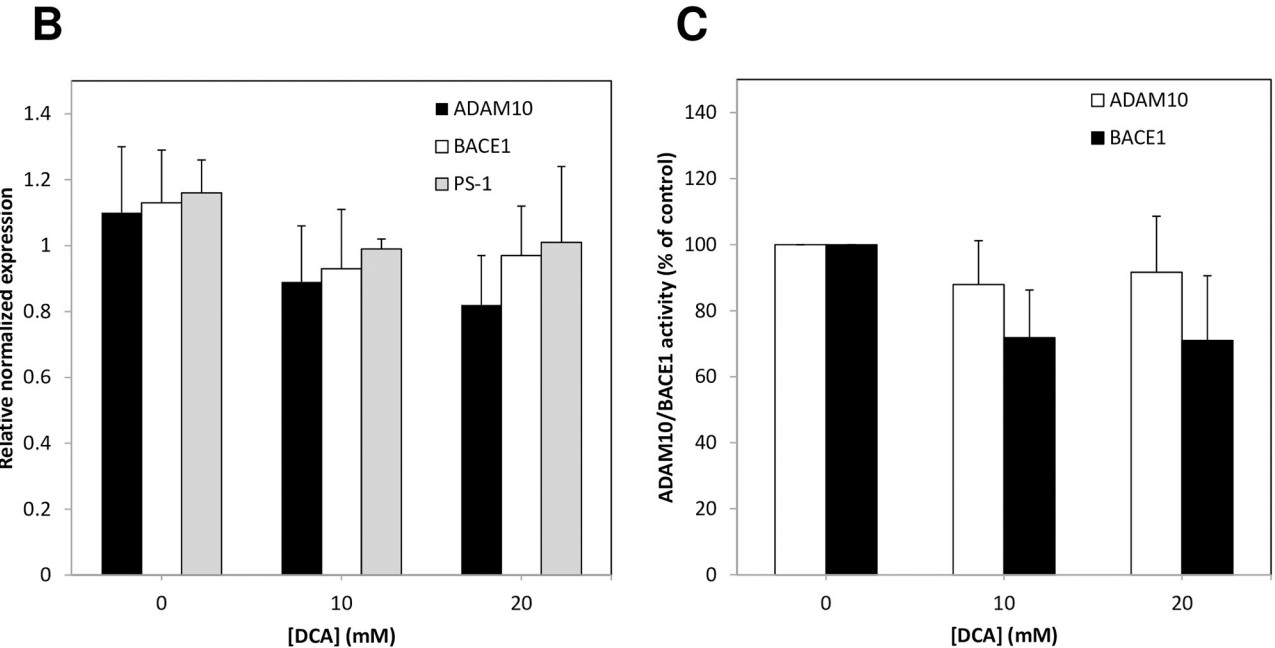

**Fig 6. DCA does not alter secretase expression/activity in untransfected SH-SY5Y cells.** Cells were grown to confluence before replacing the growth medium with UltraMEM containing the indicated DCA concentrations and culturing for an additional 24 h. Harvested cell pellets were split and used for protein and RNA extraction (Materials and methods). Parallel experiments were performed in which the resultant cell pellets were lysed and subjected to commercial fluorimetric assays for ADAM10 and BACE1 activity (see Materials and methods). **(A)** Equal amounts of protein from cell lysates were immunoblotted for ADAM10 and the blots were reprobed for actin. The positions of prodomain containing (pADAM) and mature, prodomain lacking (mADAM) forms of the enzyme are indicated. **(B)** Real-time qPCR was performed according to the Materials and methods section. Results are relative values normalized to RPLO and are means ± S.D. (n = 3). **(C)** ADAM10 and BACE1 activities in cell lysates. Results presented are non-substrate limited end-point assays determined following subtraction of ADAM10 and BACE1 inhibitor-treated control assay values (inhibitors were supplied as part of the commercial kits) and are corrected for protein levels before being expressed relative to control lysates prepared from cells incubated in the absence of DCA. Results are means ± S.D. (n = 6).

well be unintentionally overlooked if studied in such animal models. Notably, DCA also enhanced the non-amyloidogenic shedding of endogenous APP from BACE1 over-expressing cells to a similar level as it did the endogenous protein in the untransfected cells (Fig 5D). Of course, it might be argued that DCA is simply enhancing the trafficking/secretion of sAPPα.

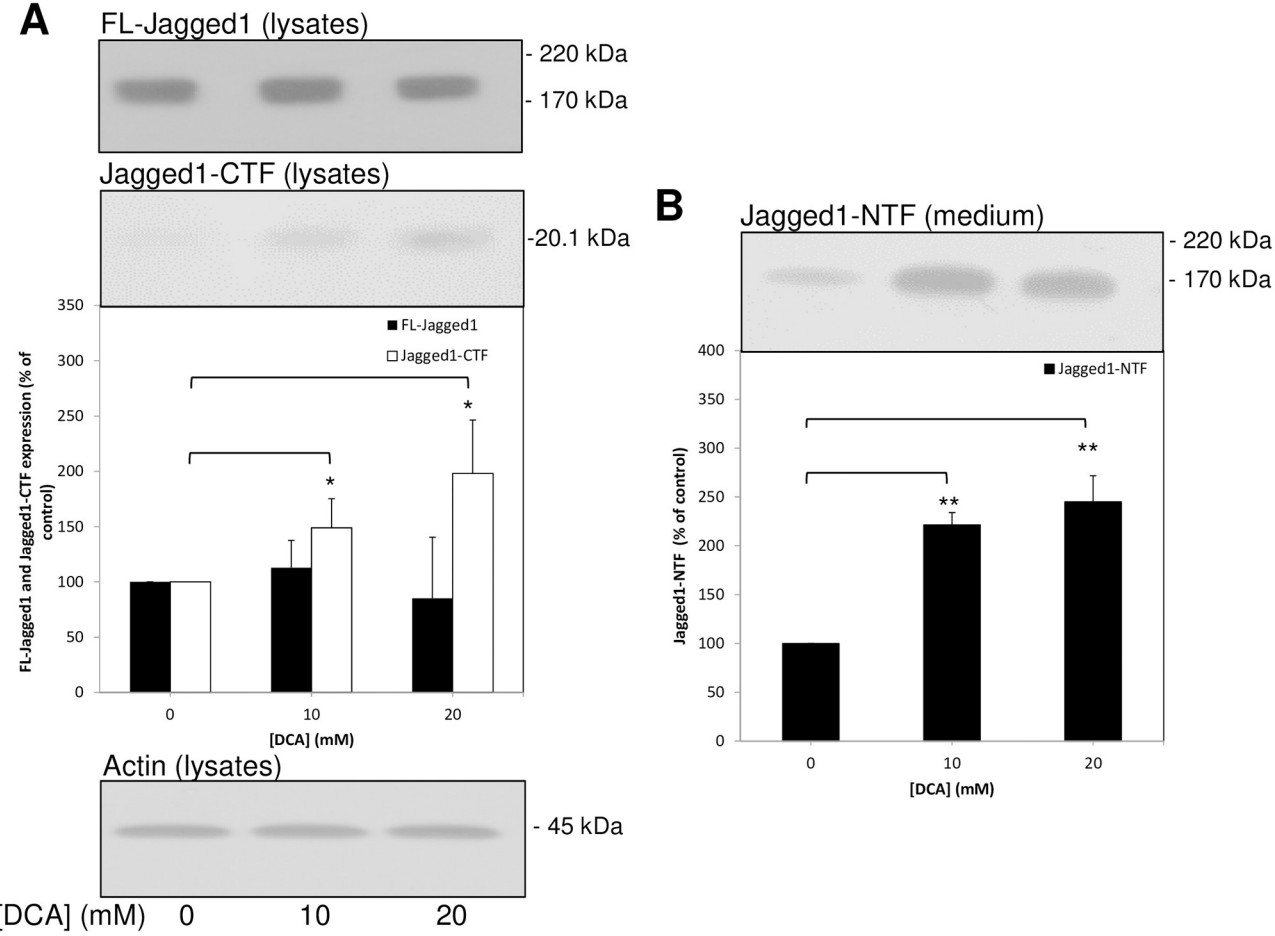

**Fig 7. DCA promotes the ADAM-mediated proteolysis of endogenous Jagged1 in SW480 cells.** Cells were grown to confluence before replacing the growth medium with UltraMEM containing the indicated DCA concentrations and culturing for an additional 24 h. Cell lysate and conditioned medium samples were prepared as described in the Materials and methods section. Equal amounts of protein (lysates) or equal volumes (medium) from samples were subjected to SDS-PAGE and immunoblotting (Materials and methods). **(A)** Cell lysates were immunoblotted with ant-Jagged1 C-terminal antibody, multiple immunoblots were the quantified in terms of both FL-Jagged1 and Jagged1 CTFs and results were expressed relative to control values. Equal protein loading was confirmed by re-probing with anti-actin antibody. **(B)** Conditioned medium samples were immunoblotted with anti-Jagged1 N-terminal antibody in order to detect the soluble Jagged1 NTF, multiples immunoblots were then quantified and results were expressed relative to control values. All results are means ± S.D. (n = 3). * = significant at $p < 0.05$; ** = significant at $p < 0.01$.

Unfortunately, the detection of this fragment in cell lysates using anti-APP antibody 6E10 is confounded by the presence of the epitope in full-length APP. However, we have also demonstrated that DCA enhances the level of α-secretase-derived C83 in cell lysates (Fig 1C) thereby proving unequivocally that non-amyloidogenic APP processing is indeed enhanced by the drug.

In terms of amyloidogenic APP processing, both sAPPβ and Aβ-peptide generation were impaired in untransfected SH-SY5Y cells treated with DCA (Figs 2 and 3) indicating that decreased β-secretase cleavage of APP was the initial event in this respect (as opposed to decreased γ-secretase cleavage of APP βCTF). This hypothesis is further supported by the fact that over-expression of BACE1 in SH-SY5Y cells ablated the effect of DCA on sAPPβ generation and, subsequently, partly ablated the effect of the drug on Aβ-peptide generation (Fig 5E and 5F). Notably this restoration of amyloidogenic processing only caused a slight reciprocal reduction of non-amyloidogenic APP processing (compare Fig 5D to Fig 3C) probably

because, even in the presence of over-expressed BACE1, only a minor proportion of the total APP pool is processed via the former pathway.

The complete mechanisms underlying the effects of DCA on APP proteolysis remain to be elucidated. However, from our results it is apparent that the drug does not simply alter the relative expression levels of amyloidogenic and non-amyloidogenic secretases nor does it impact on their cognate catalytic activities (Fig 6). However, the fact that the shedding of another secretase substrate, Jagged1, is also enhanced by DCA (Fig 7) does implicate the enhanced ADAM-mediated processing of these substrates through an, as yet, undetermined mechanism such as enhanced co-localisation of substrates and enzyme (e.g. enhanced enzyme/substrate trafficking to the cell surface or impaired re-internalisation/degradation). Note that it is currently unclear as to whether DCA enhances notch signalling as enhanced shedding of a notch ligand (Jagged1) does not necessarily translate into enhanced pathway signalling. Canonical notch signalling requires a mechanical force exerted between a receptor and ligand both attached to the surface of cells [28] and, therefore, one might, if anything, expect notch signalling to be impaired by DCA.

As to how an inhibitor of pyruvate dehydrogenase kinase might regulate any of these processes (if indeed such a mode of action is of any relevance at all in the current context) is also unclear although it is apparent that there is a connection between glucose metabolism and Alzheimer's disease. Decreased brain glucose metabolism is a well-documented event in AD as too is the accumulation of glucose and other sugars in the AD brain [29–31]. Mitochondrial dysfunction has also been documented in clinical and experimental AD studies [32] and, recently, decreased glucose oxidation in BACE1 over-expressing SH-SY5Y cells has been attributed to the impairment of tricarboxylic acid (TCA) cycle enzymes including the PDC [33]. As such, it is apparent that there is a link between increased BACE1 activity and decreased mitochondrial energy generation that can be overcome by treatment of cells with compounds capable of stimulating the TCA cycle. Whether a reciprocal mechanistic relationship between DCA-enhanced TCA activity and elevated ADAM function exists remains to be determined.

## Conclusions

In conclusion, our data show that DCA, a small molecule PDK inhibitor which traverses the blood-brain-barrier, is capable of enhancing non-amyloidogenic endogenous APP processing at a reciprocal cost to the amyloidogenic pathway and Aβ-peptide generation. These data, therefore, justify further research into the molecular mechanisms involved in these events which may lead to the development of therapies for slowing the progression of Alzheimer's disease.

## Supporting information

**S1 Fig. DCA enhances amyloidogenic processing of endogenous APP in HEK293 (A-C) and SW480 (D-F) cells.** Cells were grown to confluence before replacing the growth medium with UltraMEM containing the indicated DCA concentrations and culturing for an additional 24 h. Viability assays were then performed or cell lysate and conditioned medium samples were prepared as described in the Materials and methods section. Equal amounts of protein (lysates) or equal volumes (medium) from samples were subjected to SDS-PAGE and immunoblotting (Materials and methods). (**A**) and (**D**) Trypan blue and MTS cell viability assays. (**B** and **E**) Cell lysates were immunoblotted with anti-APP C-terminal antibody and re-probed with anti-actin antibody. Full-length APP (FL-APP) was quantified from multiple immunoblots and the results expressed relative to control values. (**C** and **F**) Conditioned medium

samples were immunoblotted with anti-APP 6E10 antibody to detect sAPPα. Multiple immunoblots were quantified and results were expressed relative to control values. All results are means ± S.D. (n = 3). * = significant at $p < 0.05$; ** = significant at $p < 0.01$; *** = significant at $p < 0.005$.
(TIF)

**S1 Raw images. Original immunoblot images.** Whole blot images are shown from which the lanes marked 'X' are excluded from cropped images shown in figures.
(TIFF)

**S1 Dataset. Minimal data set.** Means and standard deviations for individual figure panels along with the statistical testing method and levels of significance.
(DOCX)

## Acknowledgments

We would like to thank Prof. N. Hooper and Dr. N. Corbett (Faculty of Biology, Medicine and Health, University of Manchester, Manchester, United Kingdom) for their assistance with the Aβ-peptide analysis.

## Author Contributions

**Conceptualization:** Edward T. Parkin.

**Formal analysis:** Jessica E. Hammond, Matthew D. Hodges.

**Investigation:** Jessica E. Hammond, Lauren Owens.

**Methodology:** Edward T. Parkin.

**Writing – original draft:** Edward T. Parkin.

**Writing – review & editing:** Edward T. Parkin.

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
