## [Decision Letter · Decision Letter 0]

30 Sep 2021

PONE-D-21-23666The orphan drug dichloroacetate reduces amyloid beta-peptide production whilst promoting non-amyloidogenic proteolysis of the amyloid precursor proteinPLOS ONE

Dear Dr. Parkin,

Thank you for submitting your manuscript to PLOS ONE. After careful consideration, we feel that it has merit but does not fully meet PLOS ONE’s publication criteria as it currently stands. Therefore, we invite you to submit a revised version of the manuscript that addresses the points raised during the review process.

In addition to the comments raised by two Reviewers, please address the following in the revised version.

1. Although DCA did not regulate the expression of the secretases ADAM10, ADAM17, BACE1 or presenilin-1, did it increase ADAM10 enzyme activity, without increase in protein amount?,

2. The catalogue numbers of all used antibodies should be listed.

3. 10 and 20 mM final concentrations of DCA were used in this study. Is it not too high concentration?. What concentration of DCA is seen in patients receiving this compound?. Comparisons should be made and discussed.

4. sAPPα levels in Fig. 1 and sAPPβ levels in Fig. 2 as well as in other figures should be shown from lysates also in addition to the medium. Also, why was the actin shown in different panels in all the figures?. Actin should be re-probed from the same blots and should be shown under FL-APP.

5. Does DCA alter sAPPα generation from the mutant APP?. Effect on few mutations such as Swedish or Indiana should be tested and included in the revised version.

6. It is also critical to detect levels of CTFα and CTFβ using more sensitive antibodies or by concentrating the samples and the data included in the revised version.

7. If DCA also enhances notch signaling, the resulting adverse effects if any should be discussed.

We look forward to receiving your revised manuscript.

Kind regards,

Madepalli K. Lakshmana, Ph.D

Academic Editor

PLOS ONE

Additional Editor Comments (if provided):

Reviewers' comments:

Reviewer's Responses to Questions

**Comments to the Author**

1. Is the manuscript technically sound, and do the data support the conclusions?

Reviewer #1: Yes

Reviewer #2: Partly

2. Has the statistical analysis been performed appropriately and rigorously? 

Reviewer #1: Yes

Reviewer #2: Yes

3. Have the authors made all data underlying the findings in their manuscript fully available?

Reviewer #1: Yes

Reviewer #2: Yes

4. Is the manuscript presented in an intelligible fashion and written in standard English?

Reviewer #1: Yes

Reviewer #2: Yes

5. Review Comments to the Author

Reviewer #1: In this manuscript, the authors validated the ability of orphan drug dichloroacetate (DCA) to enhance non-amyloidogenic proteolysis of the amyloid precursor protein (APP) in SH-SY5Y neuroblastoma and other cell lines. However, there are some questions/suggestions listed below that would help to clarify this work.

1. Authors tried to see the CTFs on total cell lysates/CMs, but they couldn’t detect it. Why authors did not tired of Immunoprecipitate methods to detect CTFs (PMID: 32514053; PMID: 17463224).

2. There is a slight variation in cell viability of 10 um DCA treatment (Fig. 1A) between Trypan blue and MTS data. Needs clarification regarding how authors carried these two experiments; are these cells from 2 independent experiments.

3. In all the figures of Aβ quantification Aβ1-42 express relatively lower than Aβ1-40 but merging of these bars in a single y-axis scale could not able to see the differences among Aβ1-42.

4. On what basis authors selected 24 h DCA treatment, why they did not do more than 24 h.

5. In the methods sections, the authors mentioned statistical analysis was carried out either by Student’s t-test or by one-way ANOVA. But in the legend section, does not mention clearly in which data they used student’s t-test / one-way ANOVA.

Reviewer #2: This is a very intriguing study that, if verified and extended by in vivo experiments, has significant translational potential. It is disappointing, therefore, that the authors presume to know the mechanism of action of DCA in affecting the reported changes in amyloid beta-peptide production and precursor protein. In my opinion, they do not, and this is the major concern about the submission.

The presumed mechanism of DCA's effects reported here is activation of the pyruvate dehydrogenase complex by inhibition of 1 or more pyruvate dehydrogenase kinase isoforms, but this requires direct testing and validation; otherwise, the findings are largely phenomenological and not sufficiently mechanistically-oriented. Are glucose oxidation and PDC activity suppressed due to up-regulation of a PDK? Can other specific PDK inhibitors exert the same changes in amyloid metabolism as DCA? Would genetic silencing of the E1 alpha subunit render DCA ineffective under these experimental conditions? Confirming DCA's MOA would significantly enhance the probative value of the paper.

6. PLOS authors have the option to publish the peer review history of their article (what does this mean?). If published, this will include your full peer review and any attached files.

Reviewer #1: No

Reviewer #2: No

---

## [Author Response · Author response to Decision Letter 0]

15 Nov 2021

Rebuttal to editorial comments:

1. Although DCA did not regulate the expression of the secretases ADAM10, ADAM17, BACE1 or presenilin-1, did it increase ADAM10 enzyme activity, without increase in protein amount?

This is a very good question and one which we have investigated previously using fluorometric secretase assay kits in relation to both ADAM10 and BACE1 activity. In the interests of scientific openness though, we have to state that we do not subscribe entirely to the accuracy of these kits as they rely on peptides that are subject to a degree of non-specific proteolysis even when the purported ‘specific’ inhibitors are incorporated as controls as we have done (hence the reason for not incorporating these data in the original submission). However, not withstanding this caveat, we have incorporated data into Fig. 6 which show no change in either ADAM10 or BACE1 activity in the presence of DCA and added accompanying text changes in the Materials and methods, results and other necessary sections.

2. The catalogue numbers of all used antibodies should be listed.

We have now listed the relevant catalogue numbers in the ‘Materials’ subsection of the ‘Materials and methods’.

3. 10 and 20 mM final concentrations of DCA were used in this study. Is it not too high concentration?. What concentration of DCA is seen in patients receiving this compound?. Comparisons should be made and discussed.

Plasma concentrations of individuals receiving DCA therapy are approximately 10-fold lower than the 10 mM lower concentration used in the current study and we have not been able to observe any change in APP processing at a DCA concentration of 1 mM in vitro. However, it should be noted that, in vitro, higher concentrations of DCA have been shown to elicit a range of cellular effects not related to pyruvate dehydrogenase kinase (PDK) inhibition e.g. effects on mitochondrial polarisation [1] and mitophagy [2]. As such, we wanted to employ drug concentrations likely to encompass a range of mechanisms and, therefore, elected to adopt higher concentrations more commonly used in in vitro studies such as those previously published in PLOS ONE [3-6] which utilised concentrations of up to 50 mM. However, we do agree that this might limit the direct therapeutic application of DCA for the treatment of Alzheimer’s disease and have, therefore, toned down statements pertaining to this possibility in the abstract, introduction and discussion. Furthermore, we have inserted additional text at the start of the discussion pertaining to the choices made in relation to drug concentrations. We would like to emphasise that, whilst the concentrations of DCA used in the current study exceed those previously observed in plasma, this does not negate the fact that determining the mode of action of DCA with respect to APP proteolysis may lead eventually to the identification of new AD therapies.

4. sAPPα levels in Fig. 1 and sAPPβ levels in Fig. 2 as well as in other figures should be shown from lysates also in addition to the medium. Also, why was the actin shown in different panels in all the figures?. Actin should be re-probed from the same blots and should be shown under FL-APP.

We suspect that the rationale for immunoblotting lysates for sAPP� and sAPPβ may be based on the supposition that DCA may simply be affecting the secretion of these fragments? However, unfortunately, the sAPPβ antibody does not detect any of this fragment at all in lysates. Furthermore, using the anti-APP antibody 6E10 to immunoblot lysates for sAPP� is confounded by the fact that the antibody detects an epitope also present in full-length APP which is far more abundant in lysates than soluble fragments and would, therefore, mask any changes in these latter fragments.

However, we appreciate this concern but believe that the increases in APP-CTF production following DCA treatment demonstrates firmly that the drug enhances alpha-secretase-mediated proteolysis of APP rather than solely stimulating secretion of sAPPalpha (note that we have also now demonstrated that the alphaCTF/C83, specifically, is increased following drug treatment; Fig.1). Additionally, we have added this point to the Discussion of the manuscript.

Actin was reprobed after stripping blots and, therefore, is a separate exposure of the same blot. We feel that it would be a little confusing to place the actin panel immediately under the FL-APP panel as the quantification in the same panel is for APP and not the actin (e.g. Fig 1B – the quantification here correlates directly with the FL-APP blot; if the actin panel were placed between the FL-APP blot and the quantification of FL-APP it would confuse matters). By way of compromise and to make more clear that the FL-APP and actin blots are associated, we have simply combined them as a single lettered panel and altered any relevant descriptions in the results commentary where necessary. Furthermore, we have added in the blot stripping methodology to the Materials and Methods section.

5. Does DCA alter sAPPα generation from the mutant APP?. Effect on few mutations such as Swedish or Indiana should be tested and included in the revised version.

Whilst this would be interesting there are two solid reasons as to why these suggested experiments are beyond the scope of the current study or would not necessarily yield meaningful results:

(i) 95% of AD cases are sporadic and not associated with a known mutation in APP or related secretases. Our study, in examining wt-APP, is representative of this 95% of AD cases. Familial Alzheimer’s disease-associated APP mutations, in this context, are not of direct relevance.

(ii) We have shown in the current study that over-expressing APP largely ablates the observed changes in amyloidogenic/non-amyloidogenic processing that are induced by DCA in the case of endogenously expressed APP. Therefore, generating more cell lines over-expressing FAD-associated APP mutations is likely to yield artefactual results. Indeed we discuss why the use of systems over-expressing APP are not likely to be suitable for the study of AD, generally and in the current context, within the Discussion section of the current document. The only way this could be done would be to generate a series of gene-edited cell lines which is not realistic or hugely relevant in the context of the current submission. 

6. It is also critical to detect levels of CTFα and CTFβ using more sensitive antibodies or by concentrating the samples and the data included in the revised version.

See rebuttal to point 4 above. In short, we completely agree with this point as, without these data, there remains the possibility that DCA might simply be altering the secretion of APP fragments. The original CTF blots were derived from the same immunoblots as the FL-APP so are not able to tell us anything other than the fact that APP-CTFs per se increase following DCA treatment. Unfortunately, this work was done some time ago now and we no longer have access to the original samples to rerun them. However, we have now grown up untransfected SH-SY5Y cells and treated afresh with DCA subsequently preparing lysates of a higher protein concentration and resolving on Tris/tricine gels (details added to the Materials and methods section) in order to resolve individual CTF species (Fig.1). Whilst we could still not detect C99 in untransfected cells, we were able to unequivocally determine that the increase in CTFs in DCA-treated cells was specifically the consequence of C83 accumulation. We have removed the APP-CTF data from most of the other figures as, given that these samples were not run on Tris/tricine gels, the data only serves to confuse matters. We believe that our data now demonstrate unequivocally that DCA leads to an increase in alpha-secretase-mediated APP processing and does not simply alter the secretion of soluble APP fragments.

7. If DCA also enhances notch signaling, the resulting adverse effects if any should be discussed.

There is actually no evidence of this (enhanced ligand shedding does not necessarily translate into enhanced notch signalling) but it is an interesting comment. We have incorporated a very brief comment on this in the Discussion of the document.

Rebuttal to reviewer comments:

Reviewer 1:

1. Authors tried to see the CTFs on total cell lysates/CMs, but they couldn’t detect it. Why authors did not tired of Immunoprecipitate methods to detect CTFs (PMID: 32514053; PMID: 17463224).

See response to editorial comment 6.

2. There is a slight variation in cell viability of 10 um DCA treatment (Fig. 1A) between Trypan blue and MTS data. Needs clarification regarding how authors carried these two experiments; are these cells from 2 independent experiments.

Whilst the trypan blue data did show a decrease in viability at 10 mM DCA, ANOVA testing did not reveal a significant difference between the 10 mM-treated cells analysed with trypan blue and those analysed by MTS (these were, indeed, two independent experiments). We have inserted a ‘cover all’ statement at the end of the Materials and methods section pertaining to the cell viability assays which, hopefully, clarifies that these types of experiments were performed independently.

3. In all the figures of Aβ quantification Aβ1-42 express relatively lower than Aβ1-40 but merging of these bars in a single y-axis scale could not able to see the differences among Aβ1-42.

We have altered all of the Aβ quantification graphs such that they have double Y-axes. It is now much easier to see changes in Aβ1-42 levels.

4. On what basis authors selected 24 h DCA treatment, why they did not do more than 24 h.

A 24 h period was chosen as UltraMEM is a low serum medium (facilitating protein analysis by immunoblotting without the distortion of gels by large amounts of albumin derived from foetal bovine serum in complete medium) and, therefore, maintenance of cells in this medium becomes increasingly poor after 24 h. We have incorporated this explanation into the ‘Treatment of cells and protein extraction’ subsection of the Materials and methods section.

5. In the methods sections, the authors mentioned statistical analysis was carried out either by Student’s t-test or by one-way ANOVA. But in the legend section, does not mention clearly in which data they used student’s t-test / one-way ANOVA.

Actually, this is a good point – all the statistical analysis in this paper is ANOVA. We have removed the reference to student’s t-test in the Materials and methods. This was a cut and paste carry over from methods in an earlier manuscript.

Reviewer 2: 

This is a very intriguing study that, if verified and extended by in vivo experiments, has significant translational potential. It is disappointing, therefore, that the authors presume to know the mechanism of action of DCA in affecting the reported changes in amyloid beta-peptide production and precursor protein. In my opinion, they do not, and this is the major concern about the submission.

The presumed mechanism of DCA's effects reported here is activation of the pyruvate dehydrogenase complex by inhibition of 1 or more pyruvate dehydrogenase kinase isoforms, but this requires direct testing and validation; otherwise, the findings are largely phenomenological and not sufficiently mechanistically-oriented. Are glucose oxidation and PDC activity suppressed due to up-regulation of a PDK? Can other specific PDK inhibitors exert the same changes in amyloid metabolism as DCA? Would genetic silencing of the E1 alpha subunit render DCA ineffective under these experimental conditions? Confirming DCA's MOA would significantly enhance the probative value of the paper.

It is somewhat disappointing to see that Reviewer 2 answered only ‘Partly’ to the question ‘Is the manuscript technically sound, and do the data support the conclusions?’ which was, presumably, on the basis of the comments above. We cannot see how the Reviewer arrived at the conclusion that ‘the authors presume to know the mechanism of action of DCA’ or that ‘The presumed mechanism of DCA’s effects reported here is activation of the pyruvate dehydrogenase complex by 1 or more pyruvate dehydrogenase kinase isoforms…’. We actually went to considerable efforts in the writing of the manuscript to avoid these presumptions instead trying to cover several possibilities particularly in the discussion. As such, we consider these comments to have arisen from insufficient scrutiny of the manuscript and, therefore argue that, in fact, the manuscript is both technically sound and the data do indeed fully support the conclusions made. To add further to this, we have tested another PDK inhibitor which actually does not exert the same effect as DCA with respect to APP proteolysis even at concentrations known to inhibit PDK. As such, we were aiming (and think we actually achieved this) to leave options open as to the complete mechanism of DCA action whilst making informed suggestions based on our data but without extrapolating our conclusions too far. 

We have consulted the recommended templates and gone through the manuscript in detail checking for erroneous style/formatting and we think we have named submission files correctly but would be happy to make any further necessary changes. Specifically:

(i) We have capitalized proper nouns in the title.

(ii) We have indicated author affiliations with numbers only.

(iii) We have been through the entire text and ensured that symbols are inserted using the ‘Insert symbol’ function in word rather than by changing the font of existing text.

The funding body is a philanthropic organisation who donate by cheque payment to research at Lancaster University. There is no specific grant number associated with these payments.

The data are not a core part of the research as the enhanced shedding of Jagged1 is already incorporated in our figures – the comment on prion protein was merely a third example of a protein whose shedding is enhanced by DCA so the comment has been removed.

We have provided the images as requested which show the whole blots rather than cropped lanes/rows – we hope this is sufficient as it’s the only record we have of these blots but does show the whole blot area. Note that, previously, we no longer had access to the uncropped blots for ADAM17. As these blots neither showed any change following DCA treatment nor were central to the manuscript (with ADAM10 being the physiological alpha-secretase) we considered a sensible course of action here to simply remove the ADAM17 blot information from the manuscript. We have submitted our image data and minimal data set as Supporting Information.

1. Stockwin, L.H., et al., Sodium dichloroacetate selectively targets cells with defects in the mitochondrial ETC. Int J Cancer, 2010. 127(11): p. 2510-9.

2. Pajuelo-Reguera, D., et al., Dichloroacetate stimulates changes in the mitochondrial network morphology via partial mitophagy in human SH-SY5Y neuroblastoma cells. Int J Oncol, 2015. 46(6): p. 2409-18.

3. Dai, Y., et al., Dichloroacetate enhances adriamycin-induced hepatoma cell toxicity in vitro and in vivo by increasing reactive oxygen species levels. PLoS One, 2014. 9(4): p. e92962.

4. Harting, T.P., et al., Dichloroacetate affects proliferation but not apoptosis in canine mammary cell lines. PLoS One, 2017. 12(6): p. e0178744.

5. Klose, K., et al., Metformin and sodium dichloroacetate effects on proliferation, apoptosis, and metabolic activity tested alone and in combination in a canine prostate and a bladder cancer cell line. PLoS One, 2021. 16(9): p. e0257403.

6. Sun, W., et al., Chronic CSE treatment induces the growth of normal oral keratinocytes via PDK2 upregulation, increased glycolysis and HIF1alpha stabilization. PLoS One, 2011. 6(1): p. e16207.

---

## [Editor Report · Decision Letter 1]

20 Dec 2021

The orphan drug Dichloroacetate reduces Amyloid beta-peptide production whilst promoting non-amyloidogenic proteolysis of the Amyloid Precursor Protein

PONE-D-21-23666R1

Dear Dr. Parkin,

We’re pleased to inform you that your manuscript has been judged scientifically suitable for publication and will be formally accepted for publication once it meets all outstanding technical requirements.

Kind regards,

Madepalli K. Lakshmana, Ph.D

Academic Editor

PLOS ONE
---

## [Editor Report · Acceptance letter]

4 Jan 2022

PONE-D-21-23666R1 

The orphan drug Dichloroacetate reduces Amyloid beta-peptide production whilst promoting non-amyloidogenic proteolysis of the Amyloid Precursor Protein 

Dear Dr. Parkin:

I'm pleased to inform you that your manuscript has been deemed suitable for publication in PLOS ONE. Congratulations! Your manuscript is now with our production department. 

Kind regards, 

on behalf of

Dr. Madepalli K. Lakshmana 

Academic Editor

PLOS ONE